# Thorium–phosphorus triamidoamine complexes containing Th–P single- and multiple-bond interactions

Elizabeth P. Wildman[1], Gábor Balázs[2], Ashley J. Wooles[1], Manfred Scheer[2] & Stephen T. Liddle[1]

Despite the burgeoning field of uranium-ligand multiple bonds, analogous complexes involving other actinides remain scarce. For thorium, under ambient conditions only a few multiple bonds to carbon, nitrogen, oxygen, sulfur, selenium and tellurium are reported, and no multiple bonds to phosphorus are known, reflecting a general paucity of synthetic methodologies and also problems associated with stabilising these linkages at the large thorium ion. Here we report structurally authenticated examples of a parent thorium(IV)–phosphanide (Th–$PH_2$), a terminal thorium(IV)–phosphinidene (Th=PH), a parent dithorium(IV)–phosphinidiide (Th–P(H)-Th) and a discrete actinide–phosphido complex under ambient conditions (Th=P=Th). Although thorium is traditionally considered to have dominant 6d-orbital contributions to its bonding, contrasting to majority 5f-orbital character for uranium, computational analyses suggests that the bonding of thorium can be more nuanced, in terms of 5f- versus 6d-orbital composition and also significant involvement of the 7s-orbital and how this affects the balance of 5f- versus 6d-orbital bonding character.

[1] School of Chemistry, The University of Manchester, Oxford Road, Manchester M13 9PL, UK. [2] Institute of Inorganic Chemistry, University of Regensburg, Universitaets str.31, Regensburg 93053, Germany. Correspondence and requests for materials should be addressed to M.S. (email: manfred.scheer@ur.de) or to S.T.L. (email: steve.liddle@manchester.ac.uk).

n recent years the study of actinide-ligand bonding has flourished due to a desire to better understand the nature of covalency in these linkages in terms of the extent of covalency and levels of metal-derived 7s-, 7p-, 6d- and 5f-orbital character[1–3]. This desire stems not only from a need to develop our knowledge and understanding of actinide bonding at a fundamental level[4–8], but also because of potential applications in nuclear waste clean-up which might utilize extraction methods that exploit covalency differences in metal-ligand bonding[9]. Because actinide-ligand multiple bonds arguably contain the greatest opportunities to probe covalency[10–12], in recent years in addition to numerous oxo complexes[1,2] there has been intensive progress in uranium-carbene[13–19], -imido[20–28], -nitride[29–32], -phosphinidene/-arsinidene/-arsenido[33–36] and -chalcogenido (S, Se, Te) chemistry[37–40]. This work demonstrates the wide range of multiply bonded ligands that can be stabilised at uranium with appropriate supporting ligands, and also that softer, heavier main group element centres can be stabilised as well as relatively hard first-row elements such as C, N and O.

Despite the impressive, burgeoning array of uranium-ligand multiple bonds reported in recent years, analogous complexes involving other actinides remain rare[41]. This presents a challenge to the field, because arguably the most telling advances in our understanding will result from comparisons of equivalent metal-ligand linkages across different 5f-elements[42]. Thorium is an attractive target to study, because it can be handled in normal laboratories, unlike transuranic elements, but also because thorium is classically described as utilizing principally 6d-orbitals for bonding contrasting to uranium that is portrayed as more inclined to deploy 5f-orbitals[43]. However, to be verified this hypothesis requires a greater range of complexes to be prepared and compared, but progress in thorium-ligand multiple bonding pales when compared with uranium. For example, under ambient conditions there are only a handful of each class of structurally authenticated formal Th=C carbene[13,44,45], Th=N imido[46–48] and terminal Th=O oxo or Th=E•••K (E = O, S, Se, Te) complexes[49–51] known; where thorium–phosphorus multiple bonds are concerned, they are conspicuous by their absence outside of cryogenic matrix isolation experiments[52]. Indeed, of fewer than thirty crystallographically authenticated thorium–phosphorus complexes in the Cambridge Structural Database[53] seven are phosphanides[54–60], three are polyphosphides[61,62], one is a phosphinidiide[54], a term we use to indicate its bridging nature, and the remainder are phosphine derivatives; a dithorium-phosphinidiide complex [{Th($\eta^5$-C$_5$Me$_5$)$_2$(OMe)}$_2$ ($\mu$-PH)] (ref. 36) has been described though not structurally confirmed. Notably, there are no structurally authenticated thorium phosphinidenes, a term we use to indicate their terminal multiply bonded nature or -phosphidos. The paucity of thorium-ligand multiple bonds compared with uranium can be attributed to less well-developed methodologies to prepare the former, and also due to the larger (six-coordinate) effective ionic radii[63] of thorium(IV) (1.08 Å) compared with uranium(IV) (1.05 Å), which combined with the general bonding picture of thorium being more ionic, and thus labile, than uranium might require bulkier ligands to stabilize the same linkage.

Recently, we reported the synthesis of terminal parent uranium-amide/-phosphanide/-arsenide [U(Tren$^{TIPS}$)(EH$_2$)] (Tren$^{TIPS}$ = N(CH$_2$CH$_2$NSiPr$^i_3$)$_3$; E = N, P, As] and parent uranium-imido/-phosphinidene/-arsinidene [U(Tren$^{TIPS}$)(EH)] [K(L)$_2$] (L = a 15-crown-5 ether) complexes[20,33,34]. For arsenic, the arsenido tetramer [U(Tren$^{TIPS}$)AsK$_2$]$_4$ could be prepared[33], but apart from cryogenic matrix isolation conditions[52,64] there are no reports of discrete actinide-phosphidos, in-line with the rare nature of this phosphorus trianion in the f-block[65]. With uranium–pnictide complexes in-hand we have endeavoured

to prepare the thorium analogues. Here, we focus on thorium(IV)– phosphorus derivatives and report the synthesis and characterization of a terminal parent thorium(IV)–phosphanide [Th(Tren$^{TIPS}$)(PH$_2$)] (2), parent thorium(IV)–phosphinidene [Th(Tren$^{TIPS}$)(PH)][Na(12C4)$_2$] (3, 12C4 = 12-crown-4 ether), parent dithorium(IV)–phosphinidiide [{Th(Tren$^{TIPS}$)}$_2$($\mu$-PH)] (5) and dithorium(IV)-$\mu$-phosphido [{Th(Tren$^{TIPS}$)}$_2$($\mu$-P)] [Na(12C4)$_2$] (6) complexes. These complexes can be prepared by more than one route using a new thorium(IV) separated ion pair [Th(Tren$^{TIPS}$)(DME)][BPh$_4$] (1) or the thorium(IV)- cyclometallate complex [Th{N(CH$_2$CH$_2$NSiPr$^i_3$)$_2$(CH$_2$CH$_2$ NSiPr$^i_2$C(H)MeCH$_2$)}] (4) (ref. 66). Quantum chemical calculations reveal that the Th–P bonds in this study are more ionic than the U–P analogues, where comparisons are available, and in some instances show surprising levels of 7s- as well as 6d- and 5f-orbital participation in these Th–P bonds. This contrasts to the traditional focus of exclusive 6d versus 5f chemical-bonding descriptions between thorium and uranium[43], demonstrating that the bonding of thorium can be more nuanced than might be anticipated in terms of which thorium valence orbitals engage in bonding to ligands, and also how the inclusion of 7s-orbital bonding character affects the balance of 6d versus 5f character.

## Results

**Synthesis.** In our prior work[33,34] we found that salt-elimination reactions between [U(Tren$^{TIPS}$)(Cl)] and MEH$_2$ (M = Na, E = P; M = K, E = As) did not work because these pnictides are too soft to displace the hard chloride, and instead the separated ion pair complex [U(Tren$^{TIPS}$)(THF)][BPh$_4$] (ref. 34) must be utilized as the non-coordinating tetraphenylborate anion is a better leaving group than chloride. We thus identified the separated ion pair complex 1 as a suitable precursor to thorium analogues. However, initial attempts to prepare 1, by treatment of 4 (ref. 66) with Et$_3$NHBPh$_4$ in THF gave the THF-ring-opened, zwitterionic amine-stabilised primary carbocation-tethered alkoxide complex [Th(Tren$^{TIPS}$)(OCH$_2$CH$_2$CH$_2$CH$_2$NEt$_3$)][BPh$_4$] (Supplementary Figs 1–3) as the major product of this reaction (64% isolated yield). Gratifyingly, however, replacing THF solvent with DME affords 1 in 91% isolated yield (Supplementary Figs 4–6).

We find that treatment of 1 with NaPH$_2$ in DME affords the colourless parent thorium(IV)–phosphanide complex 2 in 39% crystalline yield, Fig. 1. The $^{31}$P{$^1$H} nuclear magnetic resonance (NMR) spectrum exhibits a singlet at − 144.1 p.p.m., which resonates as a triplet in the $^{31}$P NMR spectrum ($J_{PH}$ = 159 Hz) confirming the presence of the two phosphanide hydrogen atoms that resonate as a doublet at 1.4 p.p.m. in the $^1$H NMR spectrum, as determined by two-dimensional (2D)-correlation (Supplementary Figs 7–9). The attenuated total reflectance- infrared (ATR-IR) spectrum of 2 reveals P-H stretches at 2,276 and 2,251 cm$^{-1}$ (Supplementary Fig. 10) corresponding to the symmetric and antisymmetric stretches, respectively. These absorptions compare well with computed stretching frequencies for 2 of 2,305 and 2,285 cm$^{-1}$ (see below). For unequivocal confirmation, when 2 is prepared using NaPD$_2$ to give 2D the two absorptions at 2,276 and 2,251 cm$^{-1}$ shift to 1,661 and 1,639 cm$^{-1}$ (Supplementary Fig. 10), respectively (av. $\nu_{P-H}/\nu_{P-D}$ = 1.38) (ref. 36).

Treatment of 2 with benzyl sodium and two equivalents of 12C4 ether in benzene affords the yellow terminal parent thorium(IV)–phosphinidene complex 3, Fig. 1. We also find that 3 can be prepared from the cyclometallate complex 4 (ref. 65), NaPH$_2$ and two equivalents of 12C4 in 38% crystalline yield, reflecting the high solubility of this complex. The $^{31}$P{$^1$H} NMR spectrum exhibits a singlet at 150.8 p.p.m., which in the $^{31}$P NMR

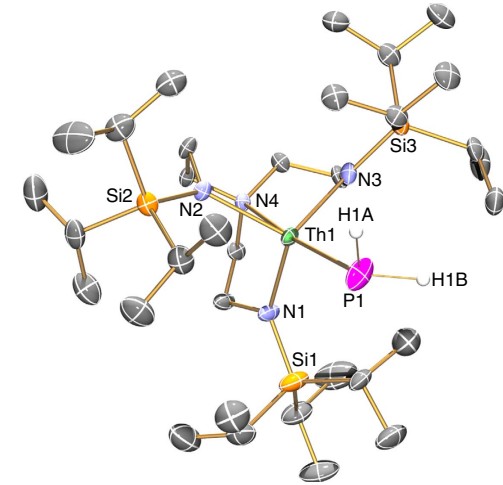

**Figure 1 | Synthetic routes to thorium–phosphorus complexes.** Complex **2** is prepared by a salt-elimination reaction of **1**. Complex **3** is prepared either by deprotonation of **2** or by a protonation reaction of the cyclometallate complex **4**. Complex **5** is prepared variously by protonation reactions involving cyclometallate **4**. Complex **6** is prepared by two different protonation strategies involving **3** or **4**.

spectrum is a doublet ($J_{PH} = 107$ Hz), confirming the presence of the phosphinidene hydrogen atom (Supplementary Figs 11 and 12); the latter resonates as a doublet at 9.4 p.p.m. in the $^1$H NMR spectrum (Supplementary Figs 13 and 14). The ATR-IR spectrum of **3** exhibits a broad absorption attributable to a P-H stretch at 2,083 cm$^{-1}$ (Supplementary Fig. 15), which compares with a computed stretching frequency for **3** of 2,095 cm$^{-1}$ (see below). When **3** is prepared using NaPD$_2$ to give **3D** the intensity of the absorbance at 2,083 cm$^{-1}$ decreases significantly and a shoulder peak at ∼1,500 cm$^{-1}$ becomes broader and more ill-defined (Supplementary Fig. 15), consistent with the presence of a broad P-D stretch ($\nu_{P\text{-}H}/\nu_{P\text{-}D} = 1.39$) (ref. 36).

With two routes to **3** established, we investigated routes to thorium–phosphido complexes and elucidated two routes to a μ-phosphido (Fig. 1). Reaction of the parent phosphanide **2** with one equivalent of cyclometallate **4** (ref. 66) effects phosphanide-deprotonation and cyclometallate-protonation to give the yellow dithorium(IV)–phosphinidiide **5**. Complex **5** is also isolated in 40% crystalline yield when NaPH$_2$ is treated with one equivalent of **4** in DME; it seems that the putative [Th(Tren$^{\text{TIPS}}$){P(H)Na}] complex reacts with NaPH$_2$ to eliminate Na$_2$PH and generate **2**, the latter of which reacts further with **4** to give **5**. The yellow dithorium(IV)-μ-phosphido complex **6** can be prepared by treating **3** with one equivalent of **4** in benzene[65], or from NaPH$_2$ treated with two equivalents of **4** (ref. 66) in the presence of two equivalents of 12C4. Complex **5** is typically obtained in yields of up to 57%. The $^{31}$P{$^1$H} NMR spectra of **5** and **6** exhibit singlets at 24.5 and 553.5 p.p.m., respectively (Supplementary Figs 16 and 17); the former is broad so in its $^{31}$P NMR spectrum the P-H coupling is not resolved and for the latter no change is observed (Supplementary Figs 18 and 19). We could not observe the PH-resonance in the $^1$H NMR spectrum of **5** even with 2D-techniques and lacking any PH-groups that of **6** is relatively uninformative (Supplementary Figs 20 and 21). The ATR-IR spectrum of **5** (Supplementary Fig. 22) reveals a broad P-H absorption at 2,116 cm$^{-1}$, and when **5** is prepared using NaPD$_2$ to give **5D** this absorption decreases significantly in intensity and a broad absorbance appears around 1,557 cm$^{-1}$ (Supplementary Fig. 22) ($\nu_{P\text{-}H}/\nu_{P\text{-}D} = 1.36$). A P-H stretching frequency of 2,216 cm$^{-1}$ is computed for **5**, likely reflecting the lack of an

anharmonic correction and the ill-defined position of the P-H vibration experimentally. The ATR-IR spectrum of **6** (Supplementary Fig. 23) is devoid of absorptions in this region consistent with the absence of any P-H groups in that complex.

**Solid-state structures.** To confirm the formulations of **2**, **3**, **5** and **6**, we determined their solid-state structures by single-crystal X-ray diffraction, Figs 2–5 (Supplementary Data 1–6). Due to the presence of heavy thorium atoms the P-H hydrogen atoms, initially located in the Fourier Transform difference map, were refined with restraints based on literature data and metrical data computed from DFT calculations (see below).

Complex **2** crystallizes with two unique molecules in the asymmetric unit but they are very similar, so only one is discussed in detail. The thorium adopts a trigonal bipyramidal geometry,

**Figure 2 | Molecular structure of 2 at 150 K with displacement ellipsoids set to 40%.** Non-phosphorus-bound hydrogen atoms and minor disorder components are omitted for clarity.

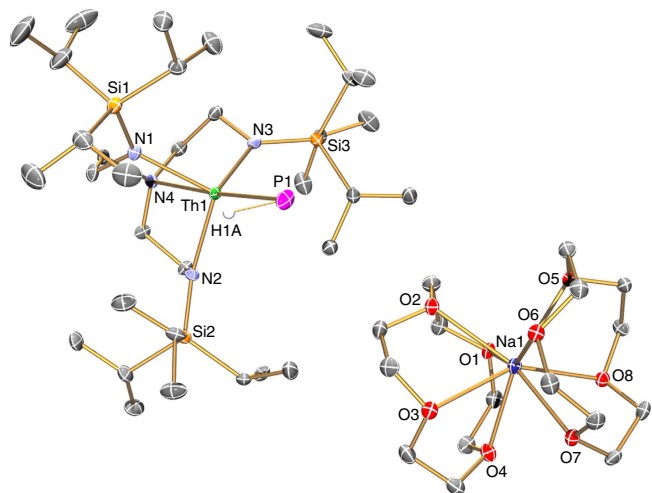

**Figure 3 | Molecular structure of 3 at 150 K with displacement ellipsoids set to 40%.** Non-phosphorus-bound hydrogen atoms are omitted for clarity. The Th···HP agostic-type interaction suggested by the structural, NMR and IR spectroscopic data is omitted for clarity.

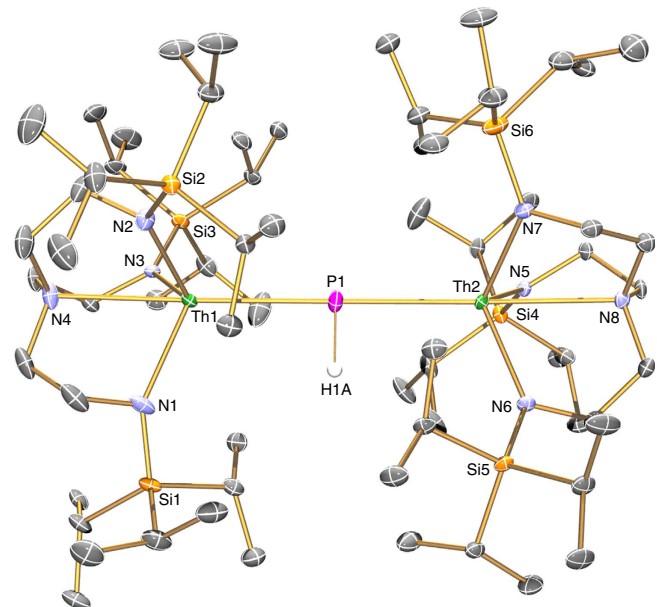

**Figure 4 | Molecular structure of 5 at 100 K with displacement ellipsoids set to 40%.** Non-phosphorus-bound hydrogen atoms, minor disorder components and lattice solvent are omitted for clarity.

where the terminal $PH_2$ unit resides *trans* to the Tren-amine centre (N4–Th1–P1 = 174.10(15)°). The Th1-P1 bond length of 2.982(2) Å is slightly longer than the sum of the single-bond covalent radii[67] for thorium and phosphorus (2.86 Å); the Th–P distance in **2** is at the upper end of thorium–phosphanide bond lengths, a feature noted with the uranium congener, and can be compared with Th–P distances in $[Th(\eta^5-C_5Me_5)_2(\mu-PPh_2)_2Ni(CO)_2]$ (av. 2.885 Å), $[Th(\eta^5-C_5Me_5)_2(\mu-PPh_2)_2PtPMe_3]$ (av. 2.923 Å), $[Th(\eta^5-C_5Me_5)_2(PPh_2)_2]$ (av. 2.866 Å), $[Th(\eta^5-C_5Me_5)_2\{P(SiMe_3)_2\}(Me)]$ (2.888(4) Å) and $[Th(\eta^5-C_5Me_5)_2(HP-2,4,6-Pr^i_3C_6H_2)_2]$ (av. 2.879 Å) (refs 54–59). The Th–P distance in **2** can also be compared with the U–P distance in the uranium analogue (2.883(2) Å) (ref. 34, the latter of which is ~0.1 Å shorter even though the single-bond covalent radius of uranium is 0.05 Å smaller than that of thorium (1.70 versus 1.75 Å, respectively)[67]. The thorium(IV)-amide and -amine distances (2.295 (av.) and 2.703(5) Å) are unexceptional[66].

The separated ion pair formulation of **3** is confirmed by the solid-state structure which shows that the phosphinidene is terminal and *trans* to the Tren$^{TIPS}$ amine centre (N4-Th1-P1 = 176.32(9)°). The Th=P double-bond length of 2.7584(18) Å is ~0.22 Å shorter than the Th–P distance in **2**, consistent with the Th=P double-bond character, and can be compared with a U=P distance in the terminal parent uranium(IV)–phosphinidene variant of **3** of 2.7159(13) Å (ref. 35), but is 0.3 Å longer than the sum of the covalent double-bond radii[67] of thorium and phosphorus (2.45 Å). The anionic nature of the thorium(IV)-containing portion of **3** is also reflected in the thorium(IV)-amide and -amine bond lengths (2.363 av. and 2.763(4) Å), which are ~0.06 Å longer than the corresponding distances in neutral **2**. The sodium cation component is unremarkable. The Th=P-H angle of 67.45(8)° suggests an agostic-type interaction between the thorium ion and the electron density of the P-H bond.

The solid-state structure of **5** reveals a dinuclear complex, and a parent phosphinidiide $HP^{2-}$ unit bridging two thorium (IV)-Tren$^{TIPS}$ fragments with essentially linear N$_{amine}$-Th–P linkages (N4–Th1–P1 = 178.71(9); N8–Th2–P1 = 177.96(10)°). The Th1-P1-Th2 angle is almost linear (177.55(6)°) and with the Th–P–H angles (89.6(5) and 89.8(5)°) presents an essentially planar T-shaped phosphinidiide centre ($\sum\angle$ = 356.95(9)°). The Th–P bond distances of 2.8982(17) and 2.8977(17) Å are statistically indistinguishable, and compare with Th–P

bond distances of 2.8083(9) and 2.8186(9) Å in the only other structurally characterized dithorium(IV)-phosphinidiide $[\{Th(\eta^5-C_5Me_5)_2\}_2\{\mu-PC_6H_2-4-Pr^i-2,6-(CH(Me)CH_2)_2\}]$ (ref. 54) and the sum of the single-bond covalent radii[67] for thorium and phosphorus (2.86 Å). The thorium(IV)-amide and -amine bond lengths (av. 2.319 and 2.755 Å, respectively) are unexceptional[66] and present metrics mid-way between those of **2** and **3**.

The dithorium(IV)-µ-phosphido formulation of **6** is revealed by the solid-state structure, and the structure of the anion component is reminiscent of the neutral ditungsten complex $[\{W(Tren^{Pr})\}_2(\mu-P)]$ [Tren$^{Pr}$ = $N(CH_2CH_2NC(H)Me_2)_3$] (ref. 68) and more broadly of the dizirconium complex $[\{Zr(\eta^5-C_5Me_5)_2\}_2(\mu-P)]$ (ref. 69). The N$_{amine}$-Th–P angles (N4-Th1-P1 = 178.16(12); N8-Th2-P1 = 175.78(12)°) are similar to **5** and the Th1-P1-Th2 angle of 176.21(8)° is almost linear. The removal of the phosphinidiide proton to generate the µ-phosphido unit is reflected by the Th1–P1 and Th2–P1 distances of 2.740(2) and 2.735(2) Å, respectively, which are 0.15 Å shorter than the Th–P distances in its formal precursor **5**, ~0.24 Å shorter than the Th–P single-bond distance in **2**, and ~0.01 Å shorter than the formal Th = P double bond in **3**. The thorium(IV)-amide and -amine bond lengths (av. 2.366 and 2.818 Å), like anionic **3**, are longer than in neutral **2** and **5**.

**Computational characterization.** To gain more insight into the nature of the Th–P bonding in **2, 3, 5** and **6** we computationally probed these molecules with DFT, NBO and QTAIM topology analyses and salient data are compiled in Table 1. Calculations were carried out on the whole molecules of **2** and **5** and the whole-anion components of **3** ($3^-$) and **6** ($6^-$). Bond lengths and angles of the geometry optimized structures (Supplementary Tables 1–4) are computed to within 0.1 Å and 3° of the experimentally derived structures so we conclude that these calculations present qualitatively useful electronic structure models.

The computed thorium MDC-$q$ charges for **2** (2.47), $3^-$ (2.24) and **5** (2.96/3.05) are in-line with thorium(IV)-Tren complexes (ref. 66), and for **3** reflect the superior donation of the terminal phosphinidene compared with phosphanide or phosphinidiide ligands. The corresponding charges for

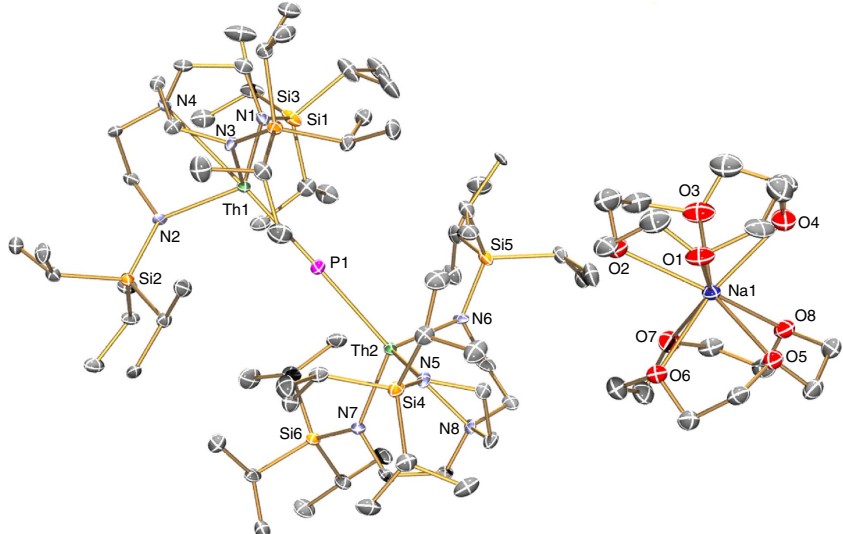

**Figure 5 | Molecular structure of 6 at 150 K with displacement ellipsoids set to 40%.** Hydrogen atoms and minor disorder components are omitted for clarity.

$6^-$ (3.56/3.46) are high and suggest ionic bonding between the two thorium ions and the phosphido. The calculated phosphorus MDC-$q$ charges reflect the changes from mono-anionic phosphanide ($-0.56$) to dianionic terminal phosphinidene ($-0.91$) to dianionic bridging phosphinidiide ($-2.26$) and trianionic phosphido ($-2.07$). The calculated Th–P Mayer bond orders of 0.77 and 1.67 for **2** and $3^-$ are consistent with formal polarized covalent Th–P single and Th=P double bonds, respectively. The analogous bond orders for **5** (0.94 and 0.95) suggest formal Th–P single bonds augmented by donation from the remaining phosphorus lone pair to each thorium, whereas the corresponding bond orders for $6^-$ (1.26 and 1.28) are consistent with the electron-rich phosphido forming Th=P pseudo double-bond interactions with each thorium but because bridging not as strongly as the terminal phosphinidene.

The Kohn–Sham orbitals of **2**, $3^-$, **5** and $6^-$, Figs 6–9, reveal that the Th–P interactions are dominated by phosphorus 3p- and a varying mix of 7s-, 5f- and 6d-orbital thorium character. The HOMO represents the principal Th–P single-bond interaction of **2** and is predominantly a phosphorus 3p-orbital that constitutes a σ-bonding interaction and pseudo lone pair. In **3**, the quasi-degenerate HOMO and HOMO−1 orbitals principally represent the thorium(IV)-phosphinidene double bond, with the former of π-symmetry, lying ∼0.02 eV higher in energy than the latter which is the σ-bond. For **5**, the HOMO is a phosphorus 3p-orbital lone pair oriented perpendicular to the Th–P–Th and P–H vectors and is partially polarized towards the two thorium centres giving weak π-interactions; the HOMO − 1 lies ∼0.67 eV below the HOMO and is a phosphorus 3p-orbital aligned along the Th–P–Th vector to give a 3-centre-2-electron σ-bonding interaction. The HOMO to HOMO − 2 of $6^-$ represent the main Th–P interactions in this compound. The HOMO and HOMO − 1 of $6^-$ are two quasi-degenerate (∼0.04 eV separation) mutually orthogonal phosphorus 3p orbitals aligned perpendicular to the Th–P–Th axis and polarized to an extent that significant π-interactions to each thorium centre are exhibited. The HOMO − 2 of $6^-$ lies ∼0.44 eV below the HOMO − 1 and like **5** represents a 3-centre-2-electron Th–P–Th σ-bonding interaction utilizing a phosphorus 3p-orbital. When taken together the two delocalized Th–P π-interactions along with the σ-bonding constitutes pseudo triple bonds to each thorium but equate to overall Th=P double bonds in a Lewis

bonding scheme, although each is not as well-developed as the phosphinidene Th=P interaction (Table 1). Although the Th–P interactions in **2**, $3^-$, **5** and $6^-$ are reasonably well isolated in their respective molecular orbital manifolds in virtually all of them intrusion of nitrogen lone pair orbital coefficients is observed. Therefore, to probe these Th–P bonding interactions more clearly we used NBO analysis.

Although the NBO calculations emphasize a more ionic bonding scenario than the DFT picture[70], they are qualitatively instructive. The σ-bond in **2** is 8% thorium character with the remaining 92% being phosphorus character. The thorium component is 12% 7s, 2% 7p, 48% 5f and 38% 6d. Reflecting the Th=P double bond in $3^-$, the σ-component is 12% thorium and 88% phosphorus 3p contributions. The thorium component to the σ-bond is distinct to that of **2**, being 4% 7s, 52% 5f and 44% 6d with no meaningful 7p contribution. The π-bond in $3^-$ shows a marginally higher thorium contribution compared with the σ-bond at 14% with the remaining 86% being phosphorus contribution. In the π-bond the thorium shows no 7s and essentially no 7p contributions (1%) with the remainder being 45% 5f and 54% 6d character. For **5** and $6^-$ ionic Th–P interactions are returned. The σ-bonds in **5** and $6^-$ return 6–8% thorium contributions with the remaining 92–94% being phosphorus character. Within the thorium contribution to the Th–P σ-bonds, the 7s contribution returns at 11–14%, but 7p contributions remain negligible at 1% and the 5f and 6d contributions are consistently 35–38 and 50–52%, respectively. For both **5** and $6^-$ the phosphorus 3p orbitals arranged orthogonally to the Th–P–Th bonds are essentially localized at phosphorus and due to the NBO detection threshold we can only say that the thorium contributes <5% to these Th–P bonds.

We examined the topologies of the Th–P bonds in **2**, $3^-$, **5** and $6^-$ using the QTAIM approach[71] to analyse the electron density $\rho(\mathbf{r})$, the Laplacian of the electron density $\nabla^2\rho(\mathbf{r})$, the electronic energy density $H(\mathbf{r})$ and the bond ellipticity parameters $\varepsilon(\mathbf{r})$. In each case 3, − 1 bond critical points (BCPs) were found indicating the presence of Th–P bonds. The BCP data with $\rho(\mathbf{r})$ values of 0.05–0.07 suggest fairly ionic bonding combinations with modest covalent contributions; covalent bonds tend to exhibit $\rho(\mathbf{r})$ values > 0.2 but whether this convention, formulated for lighter elements, is appropriate for heavy actinide elements is

**Table 1 | Experimental and calculated data for thorium-phosphorous complexes.**

| Entry[‡] | NMR $^{31}P$ $\delta$[§] | Bond length and index Th–P[‖] | BI[¶] | Atomic charges qTh[#] | qP[**] | NBO σ-component[*] % Th | % P | Th 7s:7p:5f:6d | NBO π-component[*] % Th | % P | Th 7s:7p:5f:6d | QTAIM[†] $\rho(r)$ | $\nabla^2\rho(r)$ | $H(r)$ | $\varepsilon(r)$ |
|---|---|---|---|---|---|---|---|---|---|---|---|---|---|---|---|
| **2** | −144.1 | 3.024 | 0.77 | 2.47 | −0.56 | 8 | 92 | 12:2:48:38 | — | — | — | 0.05 | 0.04 | −0.01 | 0.07 |
| **3**[−] | +150.8 | 2.709 | 1.67 | 2.24 | −0.91 | 12 | 88 | 4:0:52:44 | 14 | 86 | 0:1:45:54 | 0.07 | 0.06 | −0.02 | 0.40 |
| **5** | +24.5 | 2.994 | 0.95 | 2.96 | −2.26 | 6 | 94 | 12:1:35:52 | <5 | >95 | — | 0.05 | 0.06 | −0.01 | 0.26 |
| | — | 2.982 | 0.94 | 3.05 | — | 6 | 94 | 14:1:35:50 | <5 | >95 | — | 0.05 | 0.06 | −0.01 | 0.26 |
| **6**[−] | +553.5 | 2.768 | 1.26 | 3.56 | −2.07 | 7 | 93 | 11:1:38:50 | <5 | >95 | — | 0.06 | 0.10 | −0.02 | 0.01 |
| | — | 2.767 | 1.28 | 3.46 | — | 8 | 92 | 11:1:38:50 | <5 | >95 | — | 0.06 | 0.10 | −0.02 | 0.02 |

*Natural bond orbital analyses; the electron occupancies of these orbitals are ≥97%.
†QTAIM topological electron density ($\rho(r)$), Laplacian ($\nabla^2\rho(r)$), electronic energy density ($H(r)$) and ellipticity ($\varepsilon(r)$) bond critical point data.
‡Selected $^{31}P$ NMR spectroscopic and computed DFT, NBO and QTAIM data for **2**, the anion component of **3** (**3**[−]), **5** and the anion component of **6** (**6**[−]); all molecules geometry optimized without symmetry constraints at the restricted LDA VWN BP TZP/ZORA level.
§$^{31}P$ NMR spectroscopic chemical shift referenced relative to 85% $H_3PO_4$.
‖Calculated Th–P distances (Å).
¶Mayer bond indices.
#MDC-$q$ charges on thorium.
**MDC-$q$ charges on phosphorus.

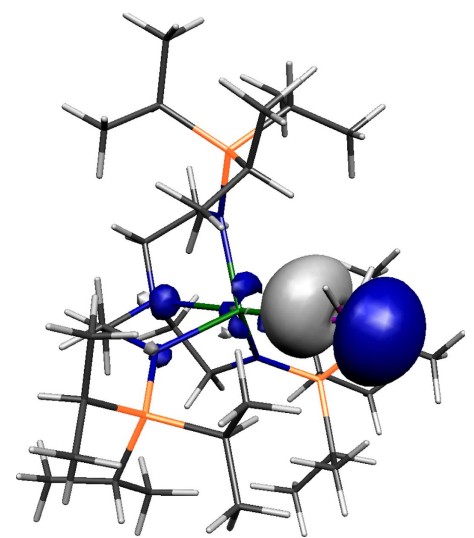

**Figure 6 | Kohn–Sham molecular orbital representation of the principal Th-P interaction of 2.** HOMO ( − 4.285 eV) represents the principal thorium–phosphorus covalent σ-bonding interaction in **2**.

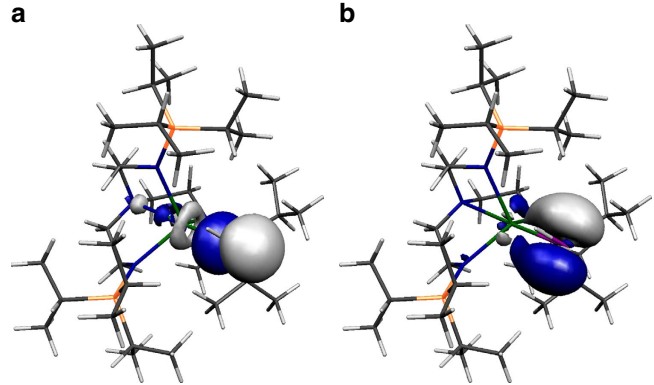

**Figure 7 | Kohn–Sham molecular orbital representations of the principal Th-P interactions of 3**[−]**.** HOMO − 1 (**a**, − 0.394 eV) and HOMO (**b**, − 0.374 eV) represent the two principal thorium–phosphorus covalent σ- and π-bonding interactions in the anion component of **3**.

debatable[33], but nevertheless notable trends emerge. The Th–P phosphanide σ-bond in **2** can be considered to present a benchmark and is fairly ionic with a spherical σ-bond ($\varepsilon(r) = 0.07$). For comparison, a σ-bond or a bond with two π-components presents a cylindrical spread of electron density ($\varepsilon(r) \sim 0$), whereas a bond with one π-component should be asymmetric ($\varepsilon(r) > 0$); for example, the $\varepsilon(r)$ parameters for the C–C bonds in ethane, benzene, and ethene are calculated to be 0.0, 0.23 and 0.45, respectively[69]. In contrast to **2**, **3**[−] exhibits an $\varepsilon(r)$ parameter of 0.40 consistent with the presence of a Th=P phosphinidene double bond, and although it is polarized it exhibits a marginally larger $\rho(r)$ value of 0.07 compared with **2**. The two $\rho(r)$ values of 0.05 for **5** are the same as for **2**, however the Th–P bonds in **5** are asymmetric with $\varepsilon(r)$ values averaging 0.26, and the QTAIM data suggest that the lone pair on the phosphinidiide may be interacting with the two thorium centres to generate asymmetric Th–P bonding combinations. The $\rho(r)$ values for **6**[−] (0.06) are intermediate to **2** and **3**[−] and the $\varepsilon(r)$ values are close to zero (0.01 and 0.02), consistent with Th–P bonds with one σ- and two π-bond compositions.

## Discussion

The difference in the (six-coordinate) effective ionic radii[63] of thorium(IV) (1.08 Å) compared with uranium(IV) (1.05 Å) is apparent even in the synthesis of the separated ion pair complex **1**. We have not observed any issues preparing [U(Tren^{TIPS})(THF)][BPh$_4$] (ref. 34) in neat THF, whereas for thorium the THF-ring-opened, zwitterionic amine-stabilised primary carbocation-tethered alkoxide complex [Th(Tren^{TIPS})(OCH$_2$CH$_2$CH$_2$CH$_2$NEt$_3$)][BPh$_4$] is the major product of the analogous preparation[66]. It would seem that compared with uranium(IV) the larger size of thorium(IV) coupled with its strong Lewis acid character is responsible for the ring-opening of THF. However, replacing THF with DME solvent circumvents this issue and provides **1** without complication.

Installation of the H$_2$P[−] parent phosphanide at thorium is relatively straightforward like the uranium analogue[33]. Surprisingly, given the prior paucity of any terminal thorium phosphinidenes, **3** can be prepared by two routes, either by deprotonation of **2** and abstraction of the sodium ion with 12C4 or by treatment of the thorium(IV)-cyclometallate complex **4** with the sodium phosphanide transfer reagent and 12C4. The latter method has the benefit that it circumvents the requirement to convert **4** to a separated ion pair followed by salt elimination, and instead deprotonation of the phosphanide, protonation of the cyclometallate, formation of a Th=P double bond and abstraction

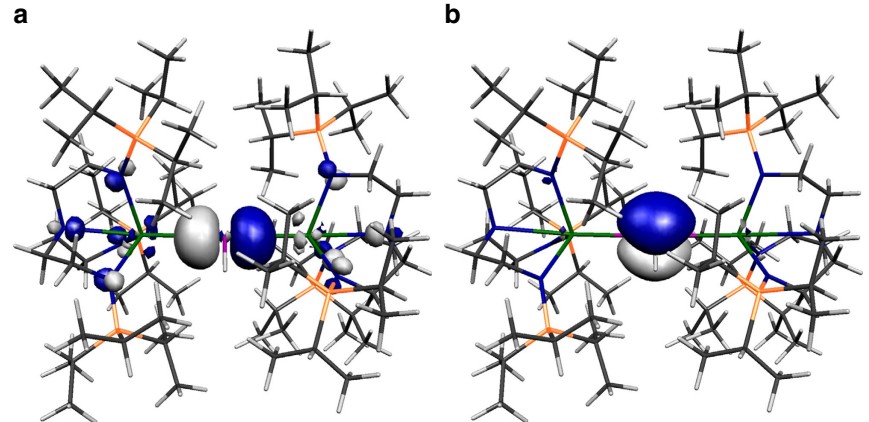

**Figure 8 | Kohn–Sham molecular orbital representations of the principal Th–P interactions of 5.** HOMO − 1 (**a**, − 4.447 eV) and HOMO (**b**, − 3.777 eV) represent the two principal thorium–phosphorus covalent σ-bonding and dative π-symmetry interactions in **5**.

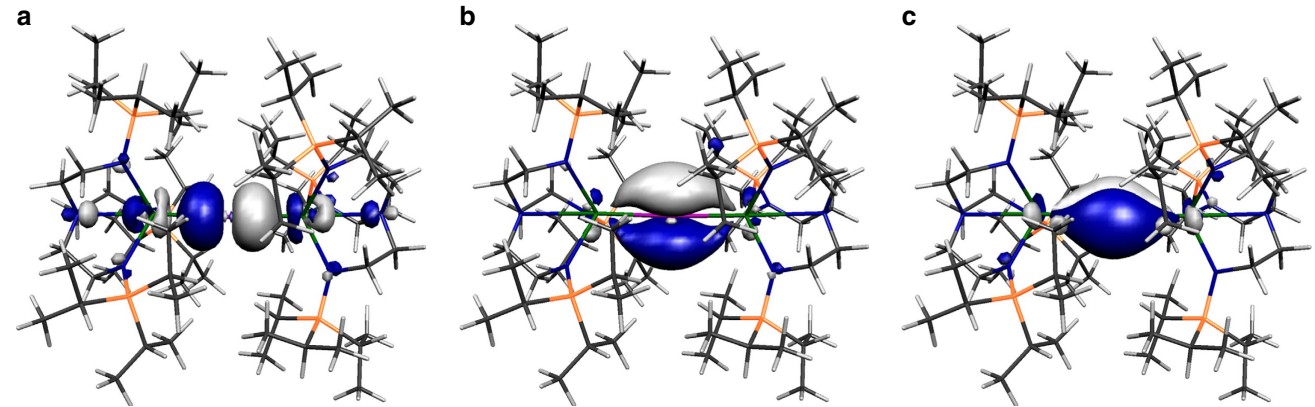

**Figure 9 | Kohn–Sham molecular orbital representations of the principal Th–P interactions of 6⁻.** HOMO − 2 (**a**, − 1.576 eV), HOMO − 1 (**b**, − 1.136 eV) and HOMO (**c**, − 1.097 eV) represent the three principal thorium–phosphorus covalent σ- and π-bonding interactions in the anion component of **6**. The two Th–P π-interactions are delocalized in the molecular orbital model but together with the σ-bond these pseudo triple bonds equate to Th = P double bonds in a Lewis bonding scheme.

of the sodium ion are all effected in one step. Whilst the preparation of **3** from **4** is notable because it works with so many bond cleavage and formation steps occurring in one reaction, we note that when **3** is prepared from **4** compound **6** is also formed in small but significant amounts (∼ 15%), presumably due to the large size of thorium coupled to the relative rates of reaction between NaPH$_2$ and **4** and once formed **3** with **4**. Germane to this point, we note that **3** is more prone to decomposition than the uranium(IV) analogue, presumably reflecting the larger size and more ionic bonding of thorium(IV) compared with uranium(IV) (see below).

A surprising aspect of the synthesis of **5** and **6**, apart from the multiple ways that they may be prepared, is the straightforward way the P–H bonds can be sequentially cleaved, leading to increasing formal charge accumulation at phosphorus. Thus, **4** easily removes a proton from **2** to give the dithorium(IV)-phosphinidiide **5**, perhaps reflecting high reactivity of the Th–C bond in **4**. More surprising, is that the dithorium(IV)-μ-phosphido **6** can be prepared directly from **4** and NaPH$_2$ with elimination of Na$_2$PH, though the reaction 2 NaPH$_2$ → Na$_2$PH + PH$_3$ has precedent.

ATR-IR data are consistent with the proposed formulations, and are generally similar to uranium analogues where comparisons are available. The P–H stretches of 2,276 and 2,251 cm$^{-1}$

for **2** compare well with 2,295 and 2,255 cm$^{-1}$ for [U(Tren$^{TIPS}$)(PH$_2$)] (ref. 35), and the P–H stretch of 2,116 cm$^{-1}$ for **5**, given its broad nature, is comparable to that of ∼ 2,193 cm$^{-1}$ for [{M(η$^5$-C$_5$Me$_5$)$_2$(OMe)}$_2$(μ-PH)] (M = U, Th)[36]. However, the P–H stretch for **3** of 2,083 cm$^{-1}$ is shifted from the P–H stretch of 2,350 cm$^{-1}$ for the uranium(IV)-phoshinidene analogue[34]. We suggest that this difference may be attributable to the different levels of covalency in the Th=PH and U=PH linkages, as evidenced by the theoretical calculations, as well as the likely Th···HP agostic-type interaction affecting the P–H bond strength, whereas the similarity of the phosphanide and phosphinidiide P–H stretches for uranium and thorium can be attributed to the largely ionic nature of the U–/Th–P interactions in these compounds. Generally, frequency calculations reproduce experimentally observed data well considering the nature of the computations, with deviations being attributable to ill-defined positions of broad absorptions and that the calculations assume a harmonic oscillator when an anharmonic model is more appropriate, but calculation of anharmonic corrections is not practicable. The $^{31}$P NMR chemical shifts of **2**, **3**, **5** and **6** (Table 1) follow the anticipated trends of shielded (**2**, − 144.1 p.p.m.), double-bond deshielded (**3**, + 150.8 p.p.m.), moderately deshielded (**5**, + 24.5 p.p.m.) and twice double-bond substantially deshielded (**6**, + 553.5 p.p.m.), consistent with their

phosphanide, phosphinidene, phosphinidiide and phosphido formulations.

The solid-state structure of **2** confirms a rare example of a terminal $H_2P^-$ phosphanide anion[34]. The Th–P bond distance in **2** is in good agreement with the sum of the single-bond covalent radii for thorium and phosphorus[67], but is relatively long for a thorium(IV)–phosphanide bond[54–59]. However, this can be rationalized on the sterically demanding Tren$^{TIPS}$ ligand whereas all other thorium(IV)–phosphanide complexes involve a *bis*(pentamethylcyclopentadienyl) ligand combination that presents a sterically more accessible wedge for phosphanide ligation[54–59]. Interestingly, the Th–P bond is $\sim 0.05$ Å longer than might be expected from the difference in covalent radii of thorium and uranium[67] when compared with the parent uranium(IV)–phosphanide congener[34], which may reflect a more ionic Th–P interaction compared with the U–P interaction as suggested by calculations (see below).

The crystal structure of **3** confirms the presence of a terminal thorium(IV)–phosphinidene linkage. On the basis of the sum of the covalent double-bond radii of thorium and phosphorus[67] the Th=P bond in **3** can be considered to be long. However, a similar effect was observed in the uranium(IV) analogue[34], and this can be attributed to the anionic, electron-rich formulation of the [Th(Tren$^{TIPS}$)(PH)]$^-$ fragment in **3** and the sterically demanding nature of Tren$^{TIPS}$. Moreover, unlike in **2** where the Th–P distance can be considered on the basis of additive covalent single-bond radii to be long when compared with the uranium analogue, in **3** the Th=P bond is $\sim 0.05$ Å longer than the corresponding U=P bond[34], but the double-bond covalent radii[67] of thorium (1.43 Å) is 0.09 Å larger than uranium (1.34 Å), and so the Th=P distance in **3** can thus be considered shorter than might have been expected on the basis of the uranium congener. Germane to this point is the acute Th=P–H bond angle, which is reminiscent of the acute M=C(R)–H angles in some alkylidenes[72] and suggests an agostic-type interaction between the thorium centre and the electron density of the P–H bond; consistent with this is the decrease of $> 50$ Hz in the $J_{PH}$ coupling moving from **2** ($J_{PH} = 159$ Hz) to **3** ($J_{PH} = 107$ Hz) as well as the infrared data, and this suggests the thorium is electron deficient and so seeks electron donation from any available electron reservoir, which may also modulate the Th=P distance.

The two Th–P bond lengths in **5** are between the Th–P distances in **2** and **3**. This can be explained on the basis that the Th–P distances in **5** should be shorter than in **2** because the $HP^{2-}$ carries a greater charge than the $H_2P^-$ unit. However, since the dianionic charge of $HP^{2-}$ is spread between two metals in **5** but only one metal in **3** it is logical that the former presents longer Th–P bonds than the latter. The Th–P distances in **5** are slightly longer than in the only other structurally authenticated dithorium(IV)–phosphinidiide complex [{Th($\eta^5$-C$_5$Me$_5$)$_2$}$_2$ {μ-PC$_6$H$_2$-4-Pr$^i$-2,6-(CH(Me)CH$_2$)$_2$}] (ref. 54), but this most likely reflects that in the latter the phosphinidiide is pinned in place by two chelating alkyl groups and also the steric encumbrance of Tren$^{TIPS}$ compared with the *bis*(pentamethylcyclopentadienyl) ligand combination. The essentially linear Th–P–Th angle in **5** contrasts to the only other structurally characterized actinide parent phosphinidiide complex [{U($\eta^5$-C$_5$Me$_5$)$_2$(OMe)}$_2$(μ-PH)][36] where the U–P–U angle is 157.7(2)°, but the energy well to bend this unit is likely quite shallow and easily driven by ligand sterics. The Th–P bond distances in **6** are shortened markedly compared with those in **5**, and indeed as well as being the first example of an actinide–phosphido complex outside of matrix isolation[52,64] are the shortest experimental Th–P bond distances on record. Although a phosphido ligand is trianionic, because it is bridging two thorium(IV) centres it would be anticipated to form at most a double-bond interaction with each thorium ion, which is reflected

in the very similar Th–P distances in **3** and **6**. However, we note that the Th–P distances in **6** are $\sim 0.29$ Å longer than the sum of the double-bond covalent radii[67] of thorium and phosphorus (2.45 Å), as is the case for **3**, but this can be justified on basis of the anionic formulation of the dithorium(IV)-μ-phosphido component of **6** and also shorter Th–P distances being precluded by the bulky Tren$^{TIPS}$ ligands that efficiently encapsulate the Th–P–Th unit. Support for these suggestions comes from inspection of the thorium(IV)–amide and –amine bond lengths which are, like for **3**, relatively long. Notably, however, even considering the anionic nature of the dithorium(IV)–phosphido unit in **6** the thorium(IV)–amine distances are very long ($> 2.8$ Å), which we suggest results from a strong *trans*-influence from the trianionic μ-phosphido.

The MDC-$q$ charges suggest that the phosphorus ligands in the complexes reported here in terms of donor ability can be ordered: $HP^{2-}$ (terminal) $> H_2P^-$ (terminal) $> HP^{2-}$ (bridging) $> P^{3-}$ (bridging) for **3**$^-$, **2**, **5** and **6**$^-$. This is consistent with what might be anticipated except for the μ-phosphido, but the bridging nature of this group clearly attenuates its donor capacity. Interestingly, calculated Mayer bond orders arrange the ligands as $HP^{2-}$ (terminal) $> P^{3-}$ (bridging) $> HP^{2-}$ (bridging) $> H_2P^-$ (terminal), as would be expected, highlighting that for thorium care must be taken when attempting to correlate calculated charges to ligand donor strength. The charge on phosphorus increases from $-0.56$ to $-0.91$ commensurate with the formally mono-anionic phosphanide and dianionic phosphinidene formulations in **2** and **3**$^-$, respectively. As expected the phosphinidiide in **5** carries a charge of $-2.26$ consistent with its dianionic formulation, and the fact that it is higher than the also dianionic phosphinidene unit in **3**$^-$ reflects that it is bound more ionically to the two thorium centres in **5** than the phosphinidene to one thorium ion in **3**$^-$. The charge on the μ-phosphido in **6**$^-$ is lower than the phosphinidiide in **5** but the μ-phosphido would be expected to be a better donor than the phosphinidiide with formal single Th–P bonds.

The DFT and NBO analyses return the anticipated Th–P bonding interactions and notably in all cases these bonding interactions return π-bonds higher in energy, even if only slightly, than σ-bonds as is normally the case. This contrasts to uranium-nitrides[31,32] where short uranium-nitrogen distances foster anti-bonding interactions that raise the energy of the uranium-nitride σ-bonds above the π-bonds, but this does not happen for the Th–P complexes here, or indeed related U–P and U–As complexes with much longer actinide–pnictide bond distances[33,34]. The consistent picture that emerges from DFT, NBO and QTAIM analyses of **2**, **3**$^-$, **5** and **6**$^-$ is that where comparisons can be made with analogous uranium(IV) complexes the thorium(IV) bonding is more ionic. For example, in [U(Tren$^{TIPS}$)(PH$_2$)] and [U(Tren$^{TIPS}$)(PH)]$^-$ (ref. 34) the bonds are composed 13% U/87% P for the U–P σ-bond for the phosphanide and 24% U/76% P for the U=P σ-bond and 28% U/72% P for the U=P π-bond of the phosphinidene, but for thorium we find 8% Th/92% P for the Th–P σ-bond of **2**, and 12% Th/88% P for the Th=P σ-bond and 14% Th/86% P for the Th=P π-bond of **3**$^-$. Thus, the thorium contributions to the metal-phosphorus bonds are consistently half that of the corresponding uranium cases. Also consistent with the more ionic bonding of thorium versus uranium is the QTAIM data[70], which whilst returning $\rho(\mathbf{r})$ values of 0.05 for both thorium and uranium parent phosphanides, shows a slight difference between thorium and uranium for the corresponding phosphinidenes (0.07 versus 0.08); although the latter data present a small change it is consistent with the overall trend. The most striking difference of the thorium contributions to the thorium–phosphorus bonds compared with the uranium

analogues is the far greater involvement of 7s-orbital contributions to the former, as high as 14%, which is in stark contrast to the latter which present at most 2% 7s-orbital contributions. Also, the 5f:6d balance is different for thorium compared with uranium. For the phosphanides the 5f-orbital contributions are essentially the same for thorium and uranium, but for thorium the 6d-orbital contribution is reduced compared with uranium because of significant 7s-orbital involvement. Where the phosphinidenes are concerned, however, the 7s- and 7p-orbital contributions are negligible for both thorium and uranium, but whereas for the former the 5f- and 6d-orbital contributions are comparable (52 versus 44%, respectively) for the latter the 5f-orbital dominates (80 versus 20%, respectively). Interestingly, for **5** and **6**$^-$, the 5f- and 6d-orbital components are reasonably balanced at ∼36 and 50%, respectively, which is perhaps surprising given that thorium bonding is traditionally considered to have dominant 6d-orbital character whereas uranium is usually considered to have dominant 5f-orbital contributions[43]. The data here suggest that the bonding of thorium can be more nuanced than might be traditionally expected and as is often observed. For example, computations comparable to the ones here on Th=O and Th=S bonds suggest larger 6d- than 5f-orbital thorium contributions to these bonds whereas for uranium analogues the latter are more involved in the U=E bonding[50], and 7s-orbital contributions are negligible. Furthermore, this nuance is not only in terms of 5f- versus 6d-orbital character but also significant involvement of the 7s-orbital and how this affects the balance of 5f- versus 6d-orbital character. On the basis of the results here it would seem that 7s-orbital participation might come at the expense of 6d- and not 5f-orbital character in thorium–phosphorus bonds.

## Methods

**General.** Experiments were carried out under a dry, oxygen-free dinitrogen atmosphere using Schlenk-line and glove-box techniques. All solvents and reagents were rigorously dried and deoxygenated before use. All compounds were characterized by elemental analyses, NMR, FTIR and UV/Vis/NIR electronic absorption spectroscopies, single-crystal X-ray diffraction studies, and DFT, NBO and QTAIM computational methods. See also Supplementary Methods.

**Preparation of [Th(Tren$^{TIPS}$)(OCH$_2$CH$_2$CH$_2$NEt$_3$)][BPh$_4$].** THF (25 ml) was added to a cold ($-78\,°C$) mixture of **4** (1.89 g, 2.24 mmol) and [HNEt$_3$][BPh$_4$] (0.94 g, 2.20 mmol). The colourless slurry was allowed to warm to room temperature and was stirred at room temperature for 16 h. The solvent and volatiles were removed *in vacuo* affording a colourless oil. The oil was washed with HMDSO (2 × 10 ml) to afford a colourless powder that was isolated by filtration and dried *in vacuo*. Yield: 2.12 g, 64%. Crystalline material suitable for X-ray crystallography was obtained from a solution in THF (3 ml) layered with hexanes (1 ml). Anal. Calcd for: C$_{67}$H$_{118}$BN$_5$OSi$_3$Th: C, 60.20; H, 8.90; N, 5.24%. Found: C, 59.09; H, 8.69; N, 4.83%. $^1$H NMR (C$_6$D$_6$, 298 K): δ 0.77 (sept, $^3J_{HH}=7.46$, 9H, C$H$(CH$_3$)$_3$), 1.00 (d, $^3J_{HH}=7.46$, 54H, CH(C$H_3$)$_3$ ), 1.36 (br, 9H, NCH$_2$C$H_3$), 2.59 (t, 6H, CH$_2$CH$_2$), 3.36 (t, 6H, CH$_2$CH$_2$), 3.52 (m, 14H, 4 × NC$H_2$CH$_3$), 7.20 (t, 7.21 Hz, 4H, p-Ar-$H$), 7.38 (t, 7.21 Hz, 8H, m-Ar-$H$), 8.08 (br, 8H, o-Ar-$H$) p.p.m. $^{29}$Si{$^1$H} NMR (C$_6$D$_6$, 298 K): not observed. $^{13}$C{$^1$H} NMR (C$_6$D$_6$, 298 K): δ 13.37 (CH(CH$_3$)$_3$), 18.81 (NCH$_2$CH$_3$), 19.60 (CH(CH$_3$)$_3$), 25.62 (NCH$_2$CH$_3$), 41.46 (CH$_2$), 45.92 (CH$_2$), 46.77 (CH$_2$), 59.86 (CH$_2$), 63.04 (CH$_2$), 68.92 (CH$_2$), 122.46 (p-Ar-CH), 126.40 (m-Ar-CH), 137.28 (o-Ar-CH) p.p.m. FTIR: v̄ 3,054 (w), 2,940 (m), 2,861 (m), 1,461 (m), 1,261 (w), 1,101 (m), 1,062 (m), 1,009 (m), 925 (m), 881 (m), 804 (w), 730 (s), 704 (s), 673 (s) 633 (m), 547 (w), 513 (w) cm$^{-1}$.

**Preparation of [Th(Tren$^{TIPS}$)(DME)][BPh$_4$] (1).** DME (25 ml) was added to a cold ($-78\,°C$) mixture of **4** (0.68 g, 0.8 mmol) and [HNEt$_3$][BPh$_4$] (0.34 g, 0.8 mmol). The colourless slurry was allowed to warm to room temperature and was stirred at room temperature for 16 h. The solution was filtered and volatiles were removed *in vacuo* affording **1** as a colourless powder. Yield: 0.85 g, 91%. Crystalline **1** as a DME solvate suitable for X-ray crystallography was obtained from a solution in C$_6$D$_6$ with a few drops of added DME stored at room temperature. Anal. Calcd for C$_{61}$H$_{105}$N$_4$BO$_2$Si$_3$Th: C, 58.44; H, 8.44; N, 4.47%. Found: C, 58.09; H, 8.40; N, 4.68%. $^1$H NMR (C$_6$D$_6$, 298 K): δ 0.79 (sept, $^3J_{HH}=7.46$, 9H, C$H$(CH$_3$)$_3$), 1.05 (d, $^3J_{HH}=7.46$, 54H, CH(C$H_3$)$_3$), 2.46 (t, 6H, CH$_2$CH$_2$), 3.01 (s, 10H, CH$_2$OC$H_3$), 3.32 (t, 6H, CH$_2$CH$_2$), 7.22 (t, $^3J_{HH}=7.21$ Hz, 4H, p-Ar-$H$), 7.39 (t, $^3J_{HH}=7.21$ Hz, 8H, m-Ar-$H$), 8.07 (br, 8H o-Ar-$H$) p.p.m.

$^{29}$Si{$^1$H} NMR (C$_6$D$_6$, 298 K): not observed. $^{13}$C{$^1$H} NMR (C$_6$D$_6$, 298 K): δ 11.84 (CH(CH$_3$)$_3$), 19.63 (CH(CH$_3$)$_3$), 41.04 (CH$_2$), 71.66 (CH$_2$), 122.30 (p-Ar-CH,), 126.18 (m-Ar-CH), 136.84, (o-Ar-CH) p.p.m. FTIR: v̄ 3,055 (w), 2,942 (m), 2,863 (m), 1,463 (m) 1,268 (w), 1,024 (m), 1,005 (m), 925 (m), 879 (m), 810 (w), 671 (m), 632 (m), 611 (m), 571 (w), 515 (w) cm$^{-1}$.

**Preparation of [Th(Tren$^{TIPS}$)(PH$_2$)] (2).** DME (25 ml) was added to a cold ($-78\,°C$) mixture of **1** (1.85 g, 1.5 mmol) and NaPH$_2$ (0.084 g, 1.5 mmol). The colourless slurry was allowed to warm to room temperature and stirred for 2.5 h to afford a pale orange solution. Solvent was removed *in vacuo* and the product was extracted into toluene. Filtration and removal of toluene *in vacuo* afforded a pale yellow solid. Crystalline **2** was obtained from a hexanes (4 ml) solution stored at room temperature. Yield: 0.50 g, 39%. Anal. Calcd for C$_{33}$H$_{77}$N$_4$PSi$_3$Th: C, 45.18; H, 8.85; N, 6.39%. Found: C, 45.46; H, 8.81; N, 6.09%. $^1$H NMR (C$_6$D$_6$, 298 K): δ 1.24 (d, $^3J_{HH}=7.21$, 54H, CH(C$H_3$)$_3$), 1.39 (sept, $^3J_{HH}=7.21$, 9H, C$H$(CH$_3$)$_3$), 1.44 (d, $J_{PH}=159.4$ Hz, 2H, P$H_2$), 2.48 (t, $^3J_{HH}=4.40$ Hz, 6H, CH$_2$CH$_2$), 3.60 (t, $^3J_{HH}=4.40$ Hz, 6H, CH$_2$CH$_2$) p.p.m. $^{31}$P NMR (C$_6$D$_6$, 298 K): δ $-144.08$ (t, $J_{PH}=159.4$ Hz, P$H_2$) p.p.m. $^{29}$Si{$^1$H} NMR (C$_6$D$_6$, 298 K): δ 5.65 (Si(CH(CH$_3$)$_2$)$_3$) p.p.m. $^{13}$C{$^1$H} NMR (C$_6$D$_6$, 298 K): δ 13.82 (CH(CH$_3$)$_2$), 20.26 (CH(CH$_3$)$_2$), 47.25 (CH$_2$), 64.02 (CH$_2$). FTIR: v̄ 2,939 (m), 2,886 (m), 2,682 (m), 2,276 (w), 2,251 (w), 1,459 (m), 1,260 (m), 1,045 (s), 1,009 (s), 923 (m), 731 (s), 672 (m), 629 (m) cm$^{-1}$. For isotopomer-shift infrared studies the P–D analogue of **2**, **2D**, was prepared by replacing NaPH$_2$ with NaPD$_2$.

**Preparation of [Th(Tren$^{TIPS}$)(PH)][Na(12C4)$_2$] (3).** Method 'A': deuterated benzene (0.5 ml) was added to an NMR tube containing **2** (29.3 mg, 3.3 × 10$^{-5}$ mol), 12C4 (11.5 mg, 6.6 × 10$^{-5}$ mol) and NaCH$_2$Ph (3.8 mg, 3.3 × 10$^{-5}$ mol). The reaction mixture was sonicated for 5 min at room temperature to afford a yellow solution. Method 'B': a solution of 12C4 (0.28 g, 1.6 mmol) in DME (25 ml) was added to a cold ($-78\,°C$) mixture of **4** (0.68 g, 0.8 mmol) and NaPH$_2$ (0.045 g, 0.8 mmol). The yellow slurry was allowed to warm to room temperature with stirring for 15 min. Solvent was immediately removed *in vacuo* to afford a bright yellow solid. The product was extracted into toluene to afford a pale yellow solution and removal of the solvent *in vacuo* afforded a yellow solid. Recrystallization of the yellow solid from toluene (4 ml) afforded yellow crystals of **3** suitable for X-ray crystallography following storage at $-30\,°C$ for 24 h. Yield: 0.38 g, 38%. Anal. Calcd for C$_{49}$H$_{108}$N$_4$PSi$_3$ThNaO$_8$● 0.5C$_7$H$_8$: C, 48.59; H, 8.70; N, 4.32%. Found: C, 48.74; H, 8.27; N, 4.64%. $^1$H NMR (C$_6$D$_6$, 298 K): δ 1.57 (d, $^3J_{HH}=7.46$, 54H, CH(C$H_3$)$_3$), 2.04 (sept, $^3J_{HH}=7.46$, 9H, C$H$(CH$_3$)$_3$), 2.61 (t, $^3J_{HH}=4.16$ Hz, 6H, CH$_2$CH$_2$), 3.30 (s, 16H, OC$H_2$), 3.72 (t, $^3J_{HH}=4.16$ Hz, 6H, CH$_2$CH$_2$), 9.44 (d, $J_{PH}=107.23$ Hz, 1H, P$H$) p.p.m. $^{31}$P NMR (C$_6$D$_6$, 298 K): δ 150.79 (d, $J_{PH}=107.59$ Hz, P$H$) p.p.m. $^{29}$Si{$^1$H} NMR (C$_6$D$_6$, 298 K): δ 2.48 (Si(CH(CH$_3$)$_3$) p.p.m. $^{13}$C{$^1$H} NMR (C$_6$D$_6$, 298 K): δ 14.69 (CH(CH$_3$)$_2$), 21.11 (CH(CH$_3$)$_2$), 46.01 (CH$_2$), 65.26 (CH$_2$), 68.08 (OCH$_2$) p.p.m. FTIR: v̄ 2,911 (m), 2,857 (m), 2,087 (w), 1,461, (w), 1,246 (m), 1,135 (s), 1,095 (s), 932 (m), 917 (m), 735 (s), 669 (m), 626 (m), 553 (m) cm$^{-1}$. For isotopomer-shift infrared studies the P–D analogue of **3**, **3D**, was prepared by method 'B' replacing NaPH$_2$ with NaPD$_2$.

**Preparation of [{Th(Tren$^{TIPS}$)}$_2$(µ-PH)] (5).** Method 'A': deuterated benzene (0.5 ml) was added to an NMR tube containing **2** (15.6 mg, 1.8 × 10$^{-5}$ mol) and **4** (15.0 mg, 1.8 × 10$^{-5}$ mol). The reaction mixture was sonicated for 5 min at room temperature to afford a yellow solution. Method 'B': DME (25 ml) was added to a cold ($-78\,°C$) mixture of **4** (0.84 g, 1 mmol) and NaPH$_2$ (0.056 g, 1 mmol). The yellow slurry was stirred at $-78\,°C$ for 35 min, allowed to warm to room temperature and stirred at room temperature for a further 1.5 h to afford a bright yellow solution. The solvent was removed *in vacuo* to afford a bright yellow solid. The product was extracted into toluene to afford a pale orange solution. Removal of the solvent *in vacuo* afforded a yellow solid. Recrystallization of the yellow solid from toluene (4 ml) afforded yellow crystals of **5** on storage at $-30\,°C$ for 24 h. The crystals were isolated and washed with pentane (3 × 3 ml) and dried *in vacuo* for 30 mins. Yield: 0.35 g, 40%. Anal. Calcd for C$_{66}$H$_{151}$N$_8$PSi$_6$Th$_2$: C, 46.07; H, 8.85; N, 6.51%. Found: C, 46.57; H, 8.91; N, 5.85%. $^1$H NMR (C$_6$D$_6$, 298 K): δ 1.42 (d, $^3J_{HH}=7.46$, 108H, CH(C$H_3$)$_3$), 1.75 (sept, $^3J_{HH}=7.46$, 18H, C$H$(CH$_3$)$_3$), 2.54 (t, $^3J_{HH}=3.91$ Hz, 12H, CH$_2$CH$_2$), 3.60 (t, $^3J_{HH}=3.91$ Hz, 12H, CH$_2$CH$_2$) p.p.m. $^{31}$P NMR (C$_6$D$_6$, 298 K): δ 24.46 (broad, µ-P$H$) p.p.m. $^{29}$Si{$^1$H} NMR (C$_6$D$_6$, 298 K): δ 3.17 (Si(CH(CH$_3$)$_3$) p.p.m. $^{13}$C{$^1$H} NMR (C$_6$D$_6$, 298 K): δ 14.41 (CH(CH$_3$)$_2$), 20.76 (CH(CH$_3$)$_2$), 46.09 (CH$_2$), 65.03 (CH$_2$) p.p.m. FTIR: v̄ 2,938 (m), 2,885 (m). 2,860 (m), 2,164 (w), 1,458 (m), 1,237 (m), 104 5 (m), 930 (s), 880 (s), 861 (m), 734 (s), 671 (s), 629 (m) cm$^{-1}$. For isotopomer-shift infrared studies the P–D analogue of **5**, **5D**, was prepared by method 'B' replacing NaPH$_2$ with NaPD$_2$.

**Preparation of [{Th(Tren$^{TIPS}$)}$_2$(µ-P)][Na(12C4)$_2$] (6).** Method 'A': deuterated benzene (0.5 ml) was added to an NMR tube containing **3** (11 mg, 8.8 × 10$^{-6}$ mol) and **4** (7.4 mg, 8.8 × 10$^{-6}$ mol). The reaction mixture was sonicated for 5 min at room temperature and yellow precipitate was observed. The solvent was removed *in vacuo* to afford a bright yellow solid. Method 'B': a solution of 12C4 (0.14 g, 0.8 mmol) and NaPH$_2$ (0.023 g, 0.4 mmol) in DME (20 ml) was added to a solution of **4** (0.68 g, 0.8 mmol) in DME (15 ml) dropwise over 15 min. The resulting yellow

solution was stirred for a further 15 min and then the solvent was removed *in vacuo* as the resulting solid was washed with toluene to afford **6** as an analytically pure bright yellow solid. Yield: 0.48 g, 57%. Crystalline **6** suitable for X-ray crystallography was obtained by dissolution in DME (5 ml) and stored at $-30\,^{\circ}\mathrm{C}$ for 24 h. Anal. Calcd for $C_{82}H_{182}N_8PSi_6Th_2NaO_8 \bullet 0.8C_7H_8$: C, 48.61; H, 8.81; N, 5.06%. Found: C, 49.05; H, 9.11; N, 5.55%. $^1$H NMR (THF-$d_8$, 298 K): $\delta$ 1.27 (d, broad, 108H, CH(CH$_3$)$_3$), 1.57 (septet, $^3J_{HH}=7.46$, 18H, C$H$(CH$_3$)$_3$), 2.53 (t, $^3J_{HH}=3.55$ Hz. 12H, C$H_2$CH$_2$), 3.68 (m, 28H, CH$_2$C$H_2$, OC$H_2$) p.p.m. $^{31}$P NMR (THF-$d_8$, 298 K): $\delta$ 553.50 (s, $\mu$-P) p.p.m. $^{29}$Si{$^1$H} NMR (THF-$d_8$, 298 K): not observed. $^{13}$C{$^1$H} NMR (THF-$d_8$, 298 K): $\delta$ 15.13 (CH(CH$_3$)$_2$), 21.62 (CH(CH$_3$)$_2$), 46.23 (CH$_2$), 59.08 (CH$_2$), 72.94 (OCH$_2$) p.p.m. FTIR: $\tilde{\nu}$ 2,913 (m), 2,855.63 (m), 2,818 (m), 1,446 (w), 1,288 (w), 1,244 (w), 994 (s), 882 (m), 796 (m), 742 (s), 655 (m), 626 (m), 556 (w), 475 (w) cm$^{-1}$.

**Preparation of NaPD$_2$.** A solution of PCl$_3$ (1.37 g, 10.0 mmol) in Et$_2$O (10 ml) was slowly added to a cold ($-90\,^{\circ}\mathrm{C}$) solution of LiAlD$_4$ (0.84 g, 20.0 mmol) in Et$_2$O (40 ml) and stirred at this temperature for 30 min. The reaction vessel was connected to a flask containing a solution of sodium anthracenide (6.40 mmol in 50 ml THF) and cooled with liquid nitrogen. The temperature of the first reaction vessel was raised to room temperature and all volatiles condensed into the flask cooled with liquid nitrogen. The liquid nitrogen bath was exchanged with a bath at $-90\,^{\circ}\mathrm{C}$ (EtOH/liquid nitrogen) and the reaction mixture stirred overnight while the temperature was allowed to rise to room temperature. The reaction mixture was concentrated to ca. 10 ml and toluene (50 ml) was added. The suspension was centrifuged and the resulting yellowish solid washed twice with warm toluene ($2 \times 40$ ml) each time. Drying the remaining solid *in vacuo* afforded a yellowish solid. Yield: 0.3 g, 78% (based on sodium anthracenide). FTIR: $\tilde{\nu}$ 1,622 (s, PD), 775 (s, PD) cm$^{-1}$ (Supplementary Figs 24 and 25).

**Data availability.** The X-ray crystallographic coordinates for structures reported in this article have been deposited at the Cambridge Crystallographic Data Centre (CCDC), under deposition number CCDC 1476593-1476598. These data can be obtained free of charge from The Cambridge Crystallographic Data Centre via www.ccdc.cam.ac.uk/data_request/cif. All other data are available from the corresponding authors on request.

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

# ARTICLE

43. Bursten, B. E., Rhodes, L. F. & Strittmatter, R. J. Bonding in tris ($\eta^5$-cyclopentadienyl) actinide complexes. 2. The ground electronic configurations of "base-free" $Cp_3An$ complexes (An = thorium, protactinium, uranium, neptunium, plutonium). *J. Am. Chem. Soc.* **111,** 2756–2758 (1989).

44. Ma, G., Ferguson, M. J. & Cavell, R. G. Actinide metals with multiple bonds to carbon: synthesis, characterization, and reactivity of U(IV) and Th(IV) bis(iminophosphorano)methandiide pincer carbene complexes. *Inorg. Chem.* **50,** 6500–6508 (2011).

45. Ren, W., Deng, X., Zi, G. & Fang, D.-C. The Th = C double bond: an experimental and computational study of thorium poly-carbene complexes. *Dalton Trans.* **40,** 9662–9664 (2011).

46. Bell, N., Maron, L. & Arnold, P. L. Thorium mono- and bis(imido) complexes made by reprotonation of *cyclo*-metalated amides. *J. Am. Chem. Soc.* **137,** 10492–10495 (2015).

47. Ren, W., Zi, G. & Walter, M. D. Synthesis, structure, and reactivity of a thorium metallocene containing a 2,2′-bipyridyl ligand. *Organometallics* **31,** 672–679 (2012).

48. Haskel, A., Straub, T. & Eisen, M. S. Organoactinide-catalyzed intermolecular hydroamination of terminal alkynes. *Organometallics* **15,** 3773–3775 (1996).

49. Smiles, D. W., Wu, G., Hrobárik, P. & Hayton, T. W. Use of $^{77}Se$ and $^{125}Te$ NMR spectroscopy to probe covalency of the actinide-chalcogen bonding in $[Th(E_n)\{N(SiMe_3)_2\}_3]^-$ (E = Se, Te; n = 1, 2) and their oxo-uranium(VI) congeners. *J. Am. Chem. Soc.* **138,** 814–825 (2016).

50. Smiles, D. E., Wu, G., Kaltsoyannis, N. & Hayton, T. W. Thorium-ligand multiple bonds *via* reductive deprotection of a trityl group. *Chem. Sci.* **6,** 3891–3899 (2015).

51. Ren, W., Zi, G., Fang, D.-C. & Walter, M. D. Thorium oxo and sulfido metallocenes: synthesis, structure, reactivity, and computational studies. *J. Am. Chem. Soc* **133,** 13183–13196 (2011).

52. Wang, X. & Andrews, L. Infrared spectra and density functional theory calculations of triplet pnictinidene $N \div ThF_3$, $P \div ThF_3$ and $As \div ThF_3$ molecules. *Dalton Trans.* **14,** 9260–9265 (2009).

53. Allen, F. H. The cambridge structural database: a quarter of a million crystal structures and rising. *Acta Cryst. Sect. B* **58,** 380 (2002).

54. Behrle, A. C., Castro, L., Maron, L. & Walensky, J. R. Formation of a bridging phosphinidene thorium complex. *J. Am. Chem. Soc.* **137,** 14846–14849 (2015).

55. Hall, S. W. *et al.* Synthesis and characterization of bis(pentamethylcyclopenta dienyl)uranium(IV) and -thorium(IV) compounds containing the bis(trimethylsilyl)phosphide ligand. *Organometallics* **12,** 752–758 (1993).

56. Edwards, P. G., Harman, M., Hursthouse, M. B. & Parry, J. S. The synthesis and crystal structure of the thorium tetraphosphido complex, $Th[P(CH_2CH_2PMe_2)_2]_4$, an actinide complex with only metal-phosphorus ligand bonds. *Chem. Commun.* 1469–1470 (1992).

57. Hay, P. J., Ryan, R. R., Salazar, K. V., Wrobleski, D. A. & Sattelberer, A. P. Synthesis and X-ray structure of $(C_5Me_5)_2Th(\mu\text{-}PPh_2)_2Pt(PMe_3)$: a complex with a thorium-platinum bond. *J. Am. Chem. Soc.* **108,** 313–315 (1986).

58. Wrobleski, D. A. *et al.* Synthesis and characterization of bis(diphenylphosphido)bis(pentamethylcyclopentadienyl)thorium(IV), $[\eta^5\text{-}C_5(CH_3)_5]_2Th(PPh_2)_2$. *Organometallics* **5,** 90–94 (1986).

59. Ritchey, J. M. *et al.* An organothorium-nickel phosphido complex with a short thorium-nickel distance. The structure of $Th(\eta^5\text{-}C_5Me_5)_2(\mu\text{-}PPh_2)_2Ni(CO)_2$. *J. Am. Chem. Soc.* **107,** 501–503 (1985).

60. Formanuik, A. *et al.* White phosphorus activation by a Th(III) complex. *Dalton Trans.* **45,** 2390–2393 (2016).

61. Behrle, A. C. & Walensky, J. R. Insertion of $^tBuNC$ into thorium-phosphorus and thorium-arsenic bonds: phosphaazaallene and arsaazaallene moieties in f element chemistry. *Dalton Trans.* **45,** 10042–10049 (2016).

62. Scherer, O. J., Werner, B., Heckmann, G. & Wolmershäuser, G. Bicyclic $P_6$ as complex ligand. *Angew. Chem. Int. Ed.* **30,** 553–555 (1991).

63. Shannon, R. D. Revised effective ionic radii and systematic studies of interatomic distances in halides and chalcogenides. *Acta Cryst. Sect. A* **32,** 751–767 (1976).

64. Andrews, L., Wang, X., Lindh, R., Roos, B. O. & Marsden, C. J. Simple $N \equiv UF_3$ and $P \equiv UF_3$ molecules with triple bonds to uranium. *Angew. Chem. Int. Ed.* **47,** 5366–5370 (2008).

65. Lv, Y., Xu, X., Chen, Y., Leng, X. & Borzov, M. V. Well-defined soluble $P^{3-}$-containing rare-earth-metal compounds. *Angew. Chem. Int. Ed.* **50,** 11227–11229 (2011).

66. Gardner, B. M. *et al.* The role of 5f-orbital participation in unexpected inversion of the $\sigma$-bond metathesis reactivity trend of triamidoamine thorium(IV) and uranium(IV) alkyls. *Chem. Sci.* **5,** 2489–2497 (2014).

67. Pyykkö, P. Additive covalent radii for single-, double-, and triply-bonded molecules and tetrahedrally bonded crystals: a summary. *J. Phys. Chem. A* **119,** 2326–2337 (2015).

68. Scheer, M., Müller, J., Schiffer, M., Baum, G. & Winter, R. Pnictides as symmetrically bridging ligands in novel neutral complexes. *Chem. Eur. J.* **6,** 1252–1257 (2000).

69. Fermin, M. C., Ho, J. & Stephan, D. W. Substituent-free $P_1$, $P_2$, and $P_3$ complexes of zirconium. *J. Am. Chem. Soc.* **116,** 6033–6034 (1994).

70. Reed, A. E., Curtiss, L. A. & Weinhold, F. Intermolecular interactions from a natural bond orbital, donor-acceptor viewpoint. *Chem. Rev.* **88,** 899–926 (1988).

71. Bader, R. F. W., Slee, T. S., Cremer, D. & Kraka, E. Descriptions of conjugation and hyperconjugation in terms of electron distributions. *J. Am. Chem. Soc.* **105,** 5061–5068 (1983).

72. Schrock, R. R. High oxidation state alkylidene and alkylidyne complexes. *Chem. Commun.* 2773–2777 (2005).

## Acknowledgements

We thank the Royal Society, European Research Council, Engineering and Physical Sciences Research Council, Universities of Nottingham, Manchester, and Regensburg, the Deutsche Forschungsgemeinschaft, and COST Action CM1006 for generously supporting this work.

## Author contributions

E.P.W. and G.B. synthesized and characterized the compounds. A.J.W. carried out the single-crystal X-ray diffraction analyses. S.T.L. performed and analysed the computational analyses, originated the central idea, and with M.S. supervised the work, analysed the data, and wrote the manuscript with contributions from all the co-authors.

## Additional information

**Competing financial interests:** The authors declare no competing financial interests.

