## [Peer review file · Nature Communications]

Reviewers' Comments:

Reviewer #1 (Remarks to the Author)

Liddle and colleagues report on the synthesis, molecular structure and DFT computations on several thorium phosphanide, phosphinidene and phosphido complexes stabilized by the sterically encumbered tripodal ligand Tren(TIPS). The paper needs to be considered in the context of the authors' previous contributions on related uranium pnictogenides [(Tren(TIPS))U(EH₂)] (E = N, P, As), [(Tren(TIPS))U(=EH)][K(L)₂] (L = 12-c-6) and [(Tren(TIPS))U(As)K₂]₄ (see refs. 20, 33, 34). The manuscript is very well-written and easy to follow. In my opinion it will be of interest to researchers working in f-element chemistry, since it addresses some fundamental questions regarding the influence of 6d- and 5f-elements to the bonding of actinide compounds (i.e. their degree of covalence). Thorium and uranium are the easiest to handle actinide elements in a regular chemical laboratory. Bonding in thorium compounds is generally believed to be more ionic (less covalent) than in uranium complexes. Furthermore, for Th the 6d-orbitals are believed to play a more dominant role than in the 5f-orbitals and this is contrary to the bonding in uranium complexes. Synthetically the thorium compounds 2 and 3 are closely related to their uranium analogues, but some modifications in the synthesis was required. A fundamental advantage of Th(IV) is also its diamagnetism, which makes ³¹P NMR a convenient tool for the characterization of compounds 3, 5 and 6. The major claim of this manuscript is that in contrast to common beliefs Th can be more subtle by incorporating also 7s-orbitals in the bonding and thereby manipulating the balance between the 5f- and 6d-orbital contribution. This is indeed a new and interesting finding!

However, although I am looking - in general - favorable on this contribution there are several issues which need to be addressed before this manuscript is actually acceptable:

1. Crystallography: There are several crystallographic problems as shown by A alerts in the checkcif file and the large number of restraints used in the refinement. However, the authors laconically state: "This is a result of the poor quality of data." Sorry to mention that: "Please grow better quality crystals!" If the authors want to discuss bond distances in detail, then the crystallography needs to be absolutely unambiguous. Nevertheless, I am not arguing with the overall connectivity.
2. The authors refer to characterization of these complexes by elemental analyses (p. 19-Materials). However, no elemental analyses are provided in the Supporting Information and I doubt based on some of their ¹H NMR spectra which contain hydrolyzed H₃tren(TIPS) as impurities that the authors can actually obtain reliable analytic data.
3. This brings me to the next point: It is not acceptable to state yields on unpurified material. What are the yields after crystallization?
4. Although not structurally authenticated Marks and co-workers also reported on [C₅Me₅]₂Th(OCH₃)]₂PH (ref. 36 in the Liddle paper), this compound has not even been mentioned in the current manuscript! This needs to be corrected and potentially also be compared computationally to the results obtained in the Liddle paper using the Tren(TIPS) ligand. Marks stated in his manuscript: "Furthermore, a doublet at δ 0.54 (J = 114.7 Hz, 1 H) is also evident and can be assigned to a P-H functionality. ... The low magnitude of this one-bond P-H coupling constant indicates the presence of unusually electropositive substituents and/or of unusual valence angles about the phosphorus atom. The ³¹P{¹H} NMR spectrum of 6 consists of a singlet at δ 74.0 that splits into a doublet (J = 114.7 Hz) in the absence of heteronuclear decoupling." Furthermore the P-H stretch of this complex was tentatively assigned to be 2193 cm⁻¹ for [C₅Me₅]₂U(OCH₃)]₂PH and 2200 cm⁻¹ for [C₅Me₅]₂U(OCH₃)]₂PH.

This is a little troublesome, because there is not significant shift between uranium and thorium. Based on this I have two additional questions:

- a) Can the authors explain the broad ^{31}P NMR resonance in compound 5? This is quite strange given the results reported by Marks. Have VT NMR studies been conducted?
- b) IR spectra: The absorptions in the IR spectra of 3 and 5 associated with the P-H stretch are actually rather circumstantial. I do not really see a strong absorption and there might as well be none. If the absorption at 2164 cm^{-1} is indeed real then there is also one absorption at ca. 3500 cm^{-1} indicating hydrolyzed H3Tren(TIPS). These IR spectra need to be recorded again to verify this claim. In the best case labels should be employed, i.e. NaPD2!

5. Page 4, line 3: Ref 66 after complex 4 is incorrect and should be ref. 65

6. Page 7, line 23-24: "agostic-type interaction" - this statement needs also to be clarified and supported by the respective characterizations: Green/Brookhart defined clear characteristics for agostic interactions and these should be applied! Is there a significant change on the $1J(\text{PH})$ coupling constant compared to related compounds?

7. Page 10, line 17: "... bonding scenario that the DFT ..." should be: "...bonding scenario than the DFT ..."

8. Some comparison to related group 4 chemistry might actually also be helpful to the reader. More out of curiosity: Would zirconium or hafnium exhibit a similar reactivity? There are some related results in the literature, e.g. Dough Stephan's $[\text{Cp}^*\text{Zr}]_2(\text{P})$ (JACS 1994,116,6033-6034).

Reviewer #2 (Remarks to the Author)

This manuscript reports the isolation of several complexes containing thorium-phosphorus bonds including the first characterization of a terminal thorium phosphinidene and bridging phosphide. The compounds are characterized through ^1H , ^{13}C , and ^{31}P NMR spectroscopy, IR spectroscopy, the structures determined by X-ray crystallography, and the electronic structure probed by quantum calculations. While the work is nearly identical to that done for their uranium system, it is well executed and worth of publication in an elite journal.

While the manuscript was submitted, an article by the Walensky group on thorium-phosphorus and -arsenic bonds was published (Dalton Trans. 2016, DOI: 10.1039/C6DT00776G). Additionally, the Walter group has a terminal thorium-oxo and sulfido (JACS 2011, 133, 13186). These should be added to the introduction.

Could the authors create a table of the ^{31}P NMR resonances for each complex? I think this would be nice to show and some discussion of the ^{31}P NMR data is welcomed.

Reviewer #3 (Remarks to the Author)

In this contribution, the authors report the synthesis of a series of new thorium complexes with single and double thorium-phosphorous bonds. Various examples can be found in the literature about actinide-ligand multiple bonding but very few involve thorium and it is the first known example of such thorium-phosphorous multiple bonds.

In addition to the experimental work, the metal-ligand bonding in the series of complexes were characterized from a detailed computational analysis. The theoretical study is well-conducted and thorium-phosphorus bonds are well described. From a NBO and QTAIM analysis, as expected, it is shown that thorium-phosphorus bonds are more ionic than uranium-phosphorus bonds in related complexes. It is suggested that $7s$ orbitals are more involved in thorium-phosphorus bonding than expected. More interestingly, according to the QTAIM analysis, double $\text{Th}=\text{P}$ bonds are only slightly more covalent than single $\text{Th}-\text{P}$ bonds even though they are significantly shorter. It would have been interesting to further investigate the bond strength in such complexes to determine to which extent covalency, bond length and strengths are related. But it may be the subject of a further

study. Finally, I have only a minor comment concerning molecular orbitals, natural localized molecular orbitals may have been a better choice instead of Kohn Sham molecular orbitals.

Reviewers' comments:

Reviewer #1 (Remarks to the Author):

Middle and colleagues report on the synthesis, molecular structure and DFT computations on several thorium phosphanide, phosphinidene and phosphido complexes stabilized by the sterically encumbered tripodal ligandm Tren(TIPS). The paper needs to be considered in the context of the authors' previous contributions on related uranium pnictogenides $[(\text{Tren}(\text{TIPS}))\text{U}(\text{EH}_2)]$ (E = N, P, As), $[(\text{Tren}(\text{TIPS}))\text{U}(\text{=EH})][\text{K}(\text{L})_2]$ (L= 12-c-6) and $[(\text{Tren}(\text{TIPS}))\text{U}(\text{As})\text{K}_2]_4$ (see refs. 20, 33, 34). The manuscript is very well-written and easy to follow. In my opinion it will be of interest to researchers working in f-element chemistry, since it addresses some fundamental questions regarding the influence of 6d- and 5f-elements to the bonding of actinide compounds (i.e. their degree of covalence). Thorium and uranium are the easiest to handle actinide elements in a regular chemical laboratory. Bonding in thorium compounds is generally believed to be more ionic (less covalent) than in uranium complexes.

Furthermore, for Th the 6d-orbitals are believed to play a more dominant role than in the 5f-orbitals and this is contrary to the bonding in uranium complexes. Synthetically the thorium compounds 2 and 3 are closely related to their uranium analogues, but some modifications in the synthesis was required. A fundamental advantage of Th(IV) is also its diamagnetism, which makes ^{31}P NMR a convenient tool for the characterization of compounds 3, 5 and 6. The major claim of this manuscript is that in contrast to common believes Th can be more subtle by incorporating also 7s-orbitals in the bonding and thereby manipulating the balance between the 5f- and 6d-orbital contribution. This is indeed a new and interesting finding!

However, although I am looking - in general - favorable on this contribution there are several issues which need to be addressed before this manuscript is actually acceptable:

1. Crystallography: There are several crystallographic problems as shown by A alerts in the checkcif file and the large number of restraints used in the refinement. However, the authors laconically state: "This is a result of the poor quality of data." Sorry to mention that: "Please grow better quality crystals!" If the authors want to discuss bond distances in detail, then the crystallography needs to be absolutely unambiguous. Nevertheless, I am not arguing with the overall connectivity.

RESPONSE: It is worth highlighting that the two data sets that suffered from poor data are 1 and the THF-ring-opened compound, and both of these compounds are peripheral to this study. We attempted several data collections but these two compounds seem to produce systematically poor crystals, but the most important point of those two structures is that the chemical connectivity and thus identity is confirmed and we don't discuss any bond distances of those complexes since we were only concerned about their identities. Importantly, the data for the central Th-P complexes are good; for those complexes the major issues were the physical limitation of the beamstop preventing reflections being detected, or absorption was causing the usual issues with heavy-atom structures, but the absorption peaks are of no chemical significance.

2. The authors refer to characterization of these complexes by elemental analyses (p. 19- Materials). However, no elemental analyses are provided in the Supporting Information and I doubt based on some of their ¹H NMR spectra which contain hydrolyzed H3tren(TIPS) as impurities that the authors can actually obtain reliable analytic data.

RESPONSE: Somehow whilst constructing the SI the relevant data were not pasted in. We sincerely apologise for the initial oversight and now include CHN data.

3. This brings me to the next point: It is not acceptable to state yields on unpurified material. What are the yields after crystallization?

RESPONSE: Usually the complexes were isolated as analytically pure compounds following pumping down and washing. However, we have now clarified the text in the SI and paper to report the actual recrystallized yields, with the exception of 6 which was obtained in acceptable purity following a straightforward wash.

4. Although not structurally authenticated Marks and co-workers also reported on [C5Me5]2Th(OCH3)]2PH (ref. 36 in the Liddle paper), this compound has not even been mentioned in the current manuscript! This needs to be corrected and potentially also be compared computationally to the results obtained in the Liddle paper using the Tren(TIPS) ligand. Marks stated in his manuscript: "Furthermore, a doublet at δ 0.54 (J = 114.7 Hz, 1 H) is also evident and can be assigned to a P-H functionality. ... The low magnitude of this one-bond P-H coupling constant indicates the presence of unusually electropositive substituents and/or of unusual valence angles about the phosphorus atom. The ³¹P{¹H} NMR spectrum of 6 consists of a singlet at δ 74.0 that splits into a doublet (J = 114.7 Hz) in the absence of heteronuclear decoupling." Furthermore the P-H stretch of this complex was tentatively assigned to be 2193 cm⁻¹ for [C5Me5]2U(OCH3)]2PH and 2200 cm⁻¹ for [C5Me5]2U(OCH3)]2PH. This is a little troublesome, because there is not significant shift between uranium and thorium.

RESPONSE: We were focused on structurally authenticated complexes but thank the reviewer for spotting this oversight and we have now acknowledged the almost certainly correct but not structurally authenticated Cp*2ThPHThCp*2 complex of Marks in the introduction and also the discussion later regarding the IR data.

Based on this I have two additional questions:

a) Can the authors explain the broad ³¹P NMR resonance in compound 5? This is quite strange given the results reported by Marks. Have VT NMR studies been conducted?

RESPONSE: We note that 5 is much larger than the Marks metallocene systems so tumbling in solution may be an issue, but there is no such problem for 6 so this may not be the case. However, the M-PH-M angles are most likely different; 5 is T-shaped, whereas the Marks system would, assuming it is isostructural to the U analogue if it ever crystallised, which seems reasonable since Marks stated the IR data were superimposable, have an angle of ca 157 °, which would have clear consequences for the NMR properties of this nucleus. VT NMR will not dramatically affect this. This is also reflected by the different ³¹P chemical shifts of 5 (24ppm) vs Marks' (74 ppm). We agree that on the face of it the superimposability of the U and Th IR data of Marks' systems seems a little odd as the reviewer points out, but we have, generally, found similar IR data for our U and Th PH2/PH complexes unless some significant difference in covalency can be invoked so there seems to be a trend there, and certainly the MPH linkage seems to be one of the more

ionic ones and thus less likely to change on going from U to Th.

b) IR spectra: The absorptions in the IR spectra of 3 and 5 associated with the P-H stretch are actually rather circumstantial. I do not really see a strong absorption and there might as well be none. If the absorption at 2164 cm⁻¹ is indeed real then there is also one absorption at ca. 3500 cm⁻¹ indicating hydrolyzed H3Tren(TIPS). These IR spectra need to be recorded again to verify this claim. In the best case labels should be employed, i.e. NaPD2!

RESPONSE: The fragility of the compounds is reflected by the fact they can be decomposed by the mechanical act of grinding, hence the occasional intrusion of tiny quantities of what is most likely TrenH3. We have collected multiple IR spectra and these are the best we can obtain (on an ATR-IR in a glove box at <0.1 ppm O2/H2O) The PH vibrations tend to be broad, however to confirm assignments we have managed to prepare non-trivial NaPD2 (synthesis and IR spectra now added to SI) and prepared 2D, 3D and 5D (IR spectra added to SI and discussion points added to manuscript) and the isotopomer shifts are as expected, thus confirming our original assignments.

5. Page 4, line 3: Ref 66 after complex 4 is incorrect and should be ref. 65

RESPONSE: This has been corrected then updated with the addition of the Walensky reference published after we submitted.

6. Page 7, line 23-24: "agostic-type interaction" - this statement needs also to be clarified and supported by the respective characterizations: Green/Brookhart defined clear characteristics for agostic interactions and these should be applied! Is there a significant change on the 1J(PH) coupling constant compared to related compounds?

RESPONSE: There are no thorium complexes that could be judged related enough with which to make the comparison, and the most relevant compounds are our uranium analogues, but since they are paramagnetic the comparison cannot be made. However, we note that the 1J(PH) decreases by over 50 Hz on moving from 2 to 3 and have now inserted a comment to that effect in the structural discussion section. The phosphinidene PH hydrogen also resonates at 9.4 ppm, which is fairly deshielded but we would stop short of attributing this to an agostic-type effect since there may also be a contribution to the chemical shift from spin orbit coupling; however this is something for another study if we can correlate 31P NMR shifts to the bonding character as we mention we would like to do below. We would also like to clarify that we deliberately and specifically use the phrase 'agostic-type', rather than just 'agostic', to accommodate the moot role of orbitals in such interactions with f elements as the reviewer is correct regarding the Green/Brookhart definitions.

7. Page 10, line 17: "... bonding scenario that the DFT ..." should be: "...bonding scenario than the DFT ..."

RESPONSE: This has been corrected.

8. Some comparison to related group 4 chemistry might actually also be helpful to the reader. More out of curiosity: Would zirconium or hafnium exhibit a similar reactivity? There are some related results in the literature, e.g. Dough Stephan's [Cp*2Zr]2(P) (JACS 1994,116,6033-6034).

RESPONSE: We have added the suggested Stephan reference when Scheer's WPW system is mentioned in the XRD section. We are now over the journal limit for references with the Walensky reference also, but the editor has kindly allowed this.

Reviewer #2 (Remarks to the Author):

This manuscript reports the isolation of several complexes containing thorium-phosphorus bonds including the first characterization of a terminal thorium phosphinidene and bridging phosphide. The compounds are characterized through 1H, 13C, and 31P NMR spectroscopy, IR spectroscopy, the structures determined by

X-ray crystallography, and the electronic structure probed by quantum calculations. While the work is nearly identical to that done for their uranium system, it is well executed and worth of publication in an elite journal. While the manuscript was submitted, an article by the Walensky group on thorium-phosphorus and -arsenic bonds was published (Dalton Trans. 2016, DOI: 10.1039/C6DT00776G). Additionally, the Walter group has a terminal thorium-oxo and sulfido (JACS 2011, 133, 13186). These should be added to the introduction. Could the authors create a table of the ^{31}P NMR resonances for each complex? I think this would be nice to show and some discussion of the ^{31}P NMR data is welcomed.

RESPONSE: We had already cited the Walter reference (#51), which after checking is confirmed as pp13183 not 13186. We have added the Walensky reference published after submission as reference #61.

Keen to avoid the manuscript becoming too long, we have inserted the ^{31}P NMR data into table 1 and then added a discussion about the ^{31}P data in the discussion section just after the IR discussion.

Reviewer #3 (Remarks to the Author):

In this contribution, the authors report the synthesis of a series of new thorium complexes with single and double thorium-phosphorous bonds. Various examples can be found in the literature about actinide-ligand multiple bonding but very few involve thorium and it is the first known example of such thorium-phosphorous multiple bonds.

In addition to the experimental work, the metal-ligand bonding in the series of complexes were characterized from a detailed computational analysis. The theoretical study is well-conducted and thorium-phosphorus bonds are well described. From a NBO and QTAIM analysis, as expected, it is shown that thorium-phosphorus bonds are more ionic than uranium-phosphorus bonds in related complexes. It is suggested that 7s orbitals are more involved in thorium-phosphorus bonding than expected. More interestingly, according to the QTAIM analysis, double Th=P bonds are only slightly more covalent than single Th-P bonds even though they are significantly shorter. It would have been interesting to further investigate the bond strength in such complexes to determine to which extent covalency, bond length and strengths are related. But it may be the subject of a further study. Finally, I have only a minor comment concerning molecular orbitals, natural localized molecular orbitals may have been a better choice instead of Kohn Sham molecular orbitals.

RESPONSE: Indeed, we are interested in seeing if we can correlate the ^{31}P NMR chemical shifts to the bonding, but feel we have to make some more compounds first to get enough data points for such comparisons to be meaningful. Regarding the last point, it's just a matter of personal taste to present the Kohn Shams, although in some ways they are more informative than the 'sanitised' NBOs so we prefer to keep the KSs.

Stephen Hill

Reviewers' Comments:

Reviewer #1 (Remarks to the Author)

The authors undertook a serious effort to address the comments/concerns raised in my original review. Overall I am very pleased with the changes they made and also the preparation of NaPD₂ in response to my question regarding the rather circumstantial assignments in the IR spectra of some of these compounds. I would personally encourage the authors to actually show the IR spectra of the P-H and P-D labeled derivatives of one compound (color-coded) in one spectrum. The absorptions are rather weak and it will be beneficial to the reader to be able to actually compare and judge the changes suggested by the authors. Since they have the raw data, this should be an easy task.

The other aspect I am a little surprised is the comment on in the reply to point 4 of my list: "The fragility of the compounds is reflected by the fact they can be decomposed by the mechanical act of grinding, ..." This is really surprising considering that the bonding is (according to their computational work) more "ionic" than for the corresponding uranium compound. Is it clear what they decompose to (besides free ligand)?

In any case with exception of my suggestion to combine the IR spectra of X-H and X-D color-coded in one Figure in the SI, I think the paper is now acceptable for Nature Communications.

Round Two Reviewer Comments

Reviewer #1 (Remarks to the Author):

The authors undertook a serious effort to address the comments/concerns raised in my original review. Overall I am very pleased with the changes they made and also the preparation of NaPD₂ in response to my question regarding the rather circumstantial assignments in the IR spectra of some of these compounds. I would personally encourage the authors to actually show the IR spectra of the P-H and P-D labeled derivatives of one compound (color-coded) in one spectrum. The absorptions are rather weak and it will be beneficial to the reader to be able to actually compare and judge the changes suggested by the authors. Since they have the raw data, this should be an easy task.

RESPONSE: We have now provided merged ATR-IR spectra of compounds with P-H linkages (2,3,5) in the SI, which has the added benefit of reducing the number of SI figures.

The other aspect I am a little surprised is the comment on in the reply to point 4 of my list: "The fragility of the compounds is reflected by the fact they can be decomposed by the mechanical act of grinding, ..." This is really surprising considering that the bonding is (according to their computational work) more "ionic" than for the corresponding uranium compound. Is it clear what they decompose to (besides free ligand)?

In any case with exception of my suggestion to combine the IR spectra of X-H and X-D color-coded in one Figure in the SI, I think the paper is now acceptable for Nature Communications.

RESPONSE: We only see a small amount of presumably free Tren, but we do not know what the final decomposition products are since they are formed on such small scales and 'in situ'.